# Shakespeare and the English Poets: The Influence of Native Speaking English Reviewers on the Acceptance of Journal Articles

**Pat Strauss** 

School of Social Sciences and Public Policy Auckland University of Technology, Private Bag 92006, Auckland 1142, New Zealand; pat.strauss@aut.ac.nz

**Abstract:** The vast majority of highly ranked academic journals use English as the means of communication. That means that academics who wish to have their research internationally recognised need to publish in English. For those who are not native speakers of English (non-anglophone), this requirement is challenging. Research indicates that these authors are at a distinct disadvantage, and that to a certain extent, this disadvantage may be exacerbated by the attitudes of reviewers. This study sought to investigate the attitudes of journal reviewers who are native speakers of English (anglophone). Eight academics who regularly review for international journals took part in semistructured interviews about their attitudes towards the kind of English they believe should be used in articles they would recommend for publication. It appears that there is a bias against language that differs from native speaker use, and that authors who employ nonstandard English might well be regarded negatively, regardless of the merits of their research. It is important, therefore, that the issue of what is regarded as appropriate English for international journals enjoys a great deal more careful consideration.

**Keywords:** English native speaking reviewers; language bias; non-traditional English

## 1. Introduction

The importance of writing for publication for academics is widely acknowledged [1–5]. McGrail et al. [1] point out that while the traditional motivation to publish because it is important to disseminate knowledge is still relevant, there are now other factors fuelling the desire to publish. Academics need to publish in order to gain tenure and promotion. Publications play an important part in gaining external research funding [2], and for some academics in countries such as Iran and China, it is a requirement that PhD candidates publish from their theses in an international journal before they graduate [2]. At the same time, it appears that it is becoming increasingly difficult for academics, particularly non-anglophone academics, to get their research published.

In 2016, over 2 million articles [6] were published in academic journals worldwide. A number of researchers claim that the rejection rates of journals are reaching record levels [7,8]. Indeed, Moustafa [9] claims that a number of highly ranked journals see a high rejection rate as a badge of honour. It is clear, therefore, that journals are overwhelmed by the sheer volume of submissions, which in turn means that journal editors are heavily reliant on reviewers to assess whether an article is worthy of publication. Clearly, where the 'publish or perish' mantra prevails, this is an important responsibility. This article explores the attitudes of anglophone reviewers towards the English employed by articles written by non-Anglophone researchers.

## 2. Literature Review

Paltridge notes ([2], p. 22) that "peer review is generally seen as the cornerstone of academic publishing", that the peer review system is viewed as the way in which journals maintain their standards because the only articles that are published are those that are considered worthy by experts working in that particular discipline area. Mulligan, Hall, and Raphael [10] echo this opinion, arguing that "the authority of peer review is so pre-eminent that the research community generally views with scepticism any research that appears in the public domain . . . which has not first been published in the peer reviewed journal" (p. 132).

There is general agreement that peer reviews have two functions. Firstly, they ensure that the articles published are considered worthy by discipline experts, and secondly, the reviews help authors to improve the quality of the written presentation ([11,12]). While these functions are laudable, it does mean that reviewers are positioned as the "gatekeepers of standards and conventions within a particular discipline" ([13], p. 49). The reviewer, therefore, is a "powerful individual" ([14], p. 396). This should not necessarily be a cause for concern. Editors choose academics to review articles because of their knowledge of and standing in the discipline.

Nonetheless, there are numerous criticisms of the system. Whether impartiality in peer evaluation is possible is a moot point, as evaluative criteria can be interpreted and applied differently [10]. There are also concerns that there are too few competent reviewers to deal with the number of journal submissions and that, as a result, less able reviewers might deliver inadequate, inconsistent or biased reports [15]. One of the characteristics of the peer review system is often viewed as a double-edged sword. Peer reviews are an occluded genre in that they are not publicly available. Generally, the reviewer has an audience of two different stakeholders—the editor and the author/s. Only the editor is aware of the reviewer's identity. The argument for the anonymity of the peer review system is a strong one. Critiquing a fellow academic's work is a face-threatening act, and if authors and reviewers know each other's identities (as is the case in some peer reviewing systems), the latter might not be as open in critiquing the proposed articles as editors might desire.

The problem, of course, is one of accountability. Concern has been expressed about the tone of reviews [12,13,16,17] with one article claiming that reviewers often forget "the basic principles of politeness that are usually observed in the academy" ([17], p. 245). Of far greater concern, however, is the fact that reviewers are not held to account for their decisions in a system where their subjectivity is recognized as a major drawback. Reviewers' impartiality cannot be accepted as a given [12].

This concern is particularly relevant in view of the widespread belief that non-anglophone authors (authors who are not native speakers of English) are prejudiced by the current system of peer reviewing [18–21]. The consequences of these non-anglophone academics losing confidence in a system described as being "at the heart of academic publishing" ([15], p. 208), are deeply concerning. If these academics are reluctant to send their research to English medium journals, "crucial knowledge" ([19], p. 148), and "fundamentally different thoughts and novel perspectives" [22] might well be lost to the wider academic community.

It must be noted that not all researchers agree that non-anglophone writers suffer discrimination. In an article that drew widespread critique, Hyland [23] argues that there is no real evidence that reviewers discriminate against non-anglophone authors. He maintains that those who argue that non-anglophone authors do suffer discrimination do not acknowledge the considerable challenges anglophone researchers face in getting published. Hyland advises non-anglophone researchers to focus on improving their language rather than positioning themselves as victims. They should not, he advises, "look for prejudice rather than revision" (p. 66). Hyland's approach has drawn criticism from a number of academics [18,20,21]. These academics point out acknowledging the difficulties that non-anglophone authors encounter does not mean that critics of Hyland's position are claiming that writing for publication is not challenging for anglophone academics. The critics are simply pointing out that it is probably more demanding for non-anglophone researchers. In response to Hyland's

argument that reviewers are not biased against non-anglophone authors, Politzer et al. note that people are "notoriously poor at identifying their own unconscious biases" ([21], p. 5).

This possible lack of impartiality appears especially important as far as the language employed in the article is concerned. In order to be recognised in their fields, academics need to publish in English [15] because in the context of academic publishing, international journals really mean English medium [3]. It is possible that concerns about how authors discuss their research might overshadow what they have to say. Bocanegra-Valle [15] claims that reviewers "are preoccupied with the linguistic infelicities in the submitted texts" (p. 214). Other research echoes the belief that reviewers focus very heavily on the language used. Gosden's research [12] indicates that referees are concerned with what he describes as the "general readability of the text" (p. 95). Belcher [24] analysed 29 reviews of papers sent to an applied linguistics journal over a three-year period. She too found that language use receives more attention than any other features of the text and that it was also the category that was most frequently commented on negatively.

It seems necessary, therefore, to investigate reviewers' attitudes towards the kind of English they believe is appropriate for use in international journals. Despite the enormous weight placed on the use of English, journals' advice to prospective authors is somewhat vague [18]. Authors are advised to use "Standard English", "clear language", "good English", and "natural English". Research suggests, however, that native speakers of English are uneasy about writing that does not mirror their own use of the language. It would appear that the kind of English the journals require is that used by English-speaking academics in countries defined as "the traditional bases of English, dominated by the mother tongue varieties of the language" ([25], p. 3). McKinley and Rose [18] claim that often, the advice given in journal guidelines to would-be authors implies that good academic English is either British or American.

Despite the pivotal role that reviewers play, it appears they do not receive training in how to review manuscripts [2,26]. In the research for his book *The Discourse of Peer Review*, Paltridge [2] interviewed 45 experienced reviewers. Over half of those to whom he spoke said that their reviewing process was based on the feedback they had received to their own articles submitted to peer reviewed journals. Others noted that it was simply something they had learnt while doing it. To exacerbate matters, it appears that here has not been a great deal of research into how reviewers actually carry out their reviews [15,16], and a number of researchers have raised the importance of analysing peer reviews [12,16]. However, I would argue that simply analysing the texts is not enough to provide a comprehensive picture of the role played by reviewers in the publication process. In an interesting article, Englander and Lopez-Bonilla [14] examined the reviewer reports of two manuscripts that had been submitted to English medium journals in the United States or Europe by non-anglophone researchers. A careful analysis of the reviews led the authors to categorise the roles of the five reviewers as:

- Ringmaster—"a person who manages the performance of others" (p. 413).
- Ally—someone who provides help by offering solutions to identified problems. An ally is eager to facilitate non-anglophone authors' participation "within the discourse community of published scientists" (p. 413).
- Guardian—sees him/herself as protector of the "community standards (being part of the in group)" (p. 413).

The current article therefore moves from analysing the text and focuses attention on the text makers. This is an important move as it draws attention to those who are writing the reviews instead of focusing simply on the texts themselves. This is in keeping with changes in attitudes towards writing. In the last 20 years, research has challenged the view that literacy (and thus writing) can be viewed as neutral and context-free [27,28]. This influential challenge to the traditional view of writing, known as the academic literacies approach, maintains that focusing solely on texts and neglecting the practices

behind these texts does not provide sufficient insight into why people write in a certain way. The approach has, as its point of departure, the concept that literacy practices are socially situated [27].

If one subscribes to this reasoning, it is important that the context in which reviewers write their reviews is explored, and not just the text that they produce. It was therefore decided that talking to reviewers about their own reviewing practices would provide a more nuanced insight into why they do what they do. The study therefore sought to investigate the attitudes of reviewers towards the kind of English they encounter in the reviewing process.

## 3. Methodology

This study was carried out at one university in New Zealand[1]. The study adopted a constructionist approach employing semistructured interviews to interact with participants. I wished to engage in dialogue with them following Koro-Ljungberg's reasoning [29] that the goal of interviews is 'to examine how knowing subjects experience or have experienced particular aspects of life' (p. 431). All the participants were native speakers of English and had at least five years' experience of reviewing for journals. The selection of participants was therefore "information-oriented" ([30], p. 307). Initially, it had been planned that there would be an equal number of participants from applied linguistics and education, but five of the interviewees work in the School of Education. As indicated earlier, all were native speakers of English and only one, a South African, was not drawn from an inner circle country. Of the other participants, six were New Zealanders and one was a Canadian. Inner circle countries are defined as traditional bases of English [24], as English is the dominant language of these countries (Table 1).

**Table 1.** (*n* = 8) Participant information.

| Code | Gender | Position | School | International/Local Reviewing Experience | Country of Origin |
|:---:|:---:|:---:|:---:|:---:|:---:|
| R1 | F | Senior lecturer | Languages | Mainly int + some local | Canada |
| R2 | M | Senior lecturer | Languages | International | New Zealand |
| R3 | M | Senior lecturer | Languages | Mainly local + some int | New Zealand |
| R4 | F | Senior lecturer | Education | Int and local | New Zealand |
| R5 | M | Senior lecturer | Education | Int and local | South Africa |
| R6 | M | Associate Professor | Education | Int and local | New Zealand |
| R7 | F | Professor | Education | Int and local | New Zealand |
| R8 | F | Senior lecturer | Education | Mainly local + some int | New Zealand |

The research was limited to two schools in one of the university's faculties. Once ethics consent had been received, the heads of the schools were approached for their permission to involve academic staff members in the project. Each head of school sent out a global email to staff in their department briefly outlining the research project and indicating that if academics were willing to participate, they should contact me directly. Eight academics indicated that they wished to participate in the research project. All had experience reviewing for a variety of journals, both local and international. All their reviewing was done in English, and for languages staff, involved articles in the area of applied linguistics. Education staff reviewed a wide range of topics related to all aspects of education. They have all published their own articles in a variety of journals. Participants R5, R6, and R7 have all served or are currently serving as journal editors.

---

[1]　I sought and was granted permission to conduct the research by the ethics committee at my own university (AUTEC 17/77). The consent was granted 23 March 2017.

The following questions were used to guide the semistructured interviews:

- Do the journals for which you review provide any specific guidelines about the language employed in the articles? If so, can you describe these requirements? How does this articulation of the requirements influence your views?
- If no guidelines are provided in this regard, how do you assess the language of the articles you review?
- A number of journals give very brief advice. For example, they note that they want Standard English/clear language/good English/natural English. How would you interpret these requirements?
- What do you think about the requirement that manuscripts of non-native speakers of English should be checked by native speakers of the language before they are submitted to journals?
- Do you comment on the language when you review an article?

These questions were sent to participants before the interview so they had time to consider their responses. The interviews, which ranged from 45 min to an hour in length, took place either in the participants' offices or my own, whichever suited their convenience. The interviews were recorded and transcribed, and then returned to the participants for checking. A number elaborated on insights given in the interviews or corrected what they felt were badly worded answers.

Open coding was initially adopted in the analysis of the data. Initially, a careful reading of the data gave rise to the following categories:

- The reviewers felt that it was the duty to help authors who were non-native speakers of English.
- The reviewers' own beliefs about the kind of English employed in the academy was very important.
- Using native speakers of English who were not discipline experts to review articles prior to submission was regarded as a questionable strategy.

However, as Gordon-Finlayson points out, "coding is simply a structure on which reflection . . . happens" ([31], p. 55). On reflection, I did not feel that I had captured the nuances of the various reviewers' positions accurately and decided to adopt Englander and Lopez-Bonilla's categories for the analysis. The data was read several times before a decision was made about the category to which a particular interviewee should be assigned. This rereading of the interviews indicated that most of the participants did not fit easily into just one category, and most moved between categories depending on circumstances.

## 4. Findings and Discussion

In this section, each interview is categorized according to Englander and Lopez-Bonilla's insights [14], and then a brief summary of the main points is given. This is followed by a discussion of the implications of the findings.

R1 raised a number of interesting issues in the interview. She clearly sees herself as a Guardian and Ringmaster and feels strongly that she is helping authors by pointing out language difficulties to them, including nontraditional use of language. She mentions "things that jump out at me whether it's a collocation of an adjective and a noun or adverb that you would sort of think oh those don't usually go together". She does not believe it is her role to suggest changes but simply to note language that she views as erroneous. She says that she herself has no difficulty with her own writing being criticized and does not believe that non-anglophone writers would take offence at her suggestions. On the contrary, she expects that they will be grateful. What was really interesting about the insights of R1 was that they went further than protection of academic English and also covered anglophone writers themselves. She noted that as an academic, she was "competing with all those people from India or around who are sending their articles in", and that there was as much pressure on her to be published as there was on them. She admitted that being a first language speaker was "a huge leg up" but that as

she had been forced to learn academic English, it is only fair that non-anglophone writers "serve an apprenticeship and publish in lower-level journals until their ability to communicate academically improves".

R2's comments place him firmly in the roles of Ringmaster and Guardian. While he is willing to indicate his concerns with the text, he does not believe that it is his role to provide suggestions or specific advice for improvement. Editors, he believes, should not send out manuscripts that are not well-written. He believes that responsibility for the language is that of the authors and notes that "if you're sending an article into a journal that is an English medium journal then you get someone who is a native speaker or an editor. You pay an editor or someone to edit your article before you send it in". In the same vein, he argues that "if those people are writing to a journal which is in a particular English language Western context then it might be that they have to adjust their way of writing". However, he believes that this is not an anglophone/non-anglophone issue, noting that some of the best academic writers he has encountered in his discipline do not have English as a first language.

R3 positions himself as both an Ally and Guardian. Although he claims that he is quite flexible in his acceptance of different language, he notes that he does find it difficult to accept non-standard English. He is sympathetic towards non-anglophone authors but cannot ignore errors, putting this down to his background as an English language teacher. He believes that in drawing an author's attention to nonstandard use of English, he is doing them a service. "Personally I think there is a point where you can't flaunt these linguistic features [syntax collocation, use of tenses] without compromising meaning". He is prepared to offer help to authors but believes that journals not only have a right but also an obligation to maintain linguistic standards. "I don't think it would get a manager in a five-star hotel wanting to lower the standards to a four-star hotel so if you take a prestigious journal one of the criteria I am sure is applied concerns language standards". However, he acknowledges that his judgement is probably affected by the fact that English is his first language: "and here is the rub. Perhaps I do have a bias related to my nativeness".

R4 appears to be the reviewer who would be categorised most strongly as an Ally. She has never rejected a journal article and believes it is the duty of reviewers to ensure that people's voices are heard. She has offered her services to mentor authors who may be struggling to get published. However, even she is influenced by pragmatic concerns. She likes to make use of Track Changes and thus prefers to work in Word documents and acknowledges that she will give less advice if she is not familiar with the format of the article. Her efforts to assist non-anglophone authors would not, however, extend to challenging the current system, even though she believes that it unfairly advantages anglophone authors. Instead, she believes that it is her job to help authors with the game "the game of getting published in academic journals, playing by the rules, proper English, is one of the rules". She believes the current system dominated by anglophone academics is too powerful to be challenged.

R5 clearly sees his role as that of Guardian. In the interview, he referred repeatedly to a textbook about the style of writing written 60 years ago. He notes that he takes great pride in his own writing and says that when the articles he reviews do not meet his standards, he becomes "quite irritated". He discourages where he can the use of American English: "for me that's the standard, Shakespeare and English poets".

R6 moves between the role of Ally and Ringmaster depending on the time available to him, and possibly also how engaged he is with the text. He understands the role of Guardian and acknowledges that editors are concerned about maintaining standards for the journals for which they are responsible but believes that there should be "affirmative action" on journal publication. He would like to see the establishment of journals which accommodate a wide range of academic English as he believed that this would make it easier for editors and peer reviewers to be more accommodating of different styles of writing because "as an editor and as a peer reviewer you worry about the status, the reputation of the journal, the way it will be received". This interviewee was the only one to raise concerns about non-anglophone readers apprehensive that if a journal was more flexible in the English it accepted, this might be confusing for these readers:

> *There will be people reading this who are struggling with the English language or who have a different understanding of English language. We probably imagine that the reader is really confident with the English language and therefore can get through some of the grammatical errors but if the writer's first language is let's say Mandarin and the reader's first language is let's say Finnish then do I have a responsibility as the reviewer to support expression that works across that domain?*

R7 appears to move between all three roles. If there is only the occasional lapse in the articles she is reviewing, she will list the errors and suggest ways to correct them because she thinks that "the quality of the thinking outweighs the problematics of the expression". However, more substantial errors even in an article which she thinks has merit will be returned with a suggestion that the author engage the services of a proof reader. She maintains that she is only interested in errors and tolerates nontraditional language, that she does not try "to turn second language into first language English". However, she concedes that for her, good English is "whatever I happen to like", and that she is irritated with authors who appear to have a limited vocabulary. She feels that in some ways, non-anglophone writing is better because authors "often get to the point because they haven't got the facility to beat around the bush the way that English writers do". This seems to be somewhat of a contradiction. On the one hand, she admires the succinctness of non-anglophone writing, but on the other, she decries the lack of vocabulary resources she believes leads to this succinctness. She adds although she is prepared to accept "a different kind of English", she warns that her tolerance should not be tested.

R8, who is clearly an Ally, says on the whole, as long as the meaning is clear, she does not suggest changes. She notes that where "the sentences are quite wobbly" and interfere with meaning, she will suggest alterations, otherwise "I think it is quite nice for people to read something that is not necessarily written exactly the same way as I would write it". She might suggest that authors ask for help from someone "who is a stronger speaker of English" to help with revisions, but she would not turn down an article because of the language, noting that it was the substance that interested her. She is prepared to invest time and effort into explanations that could be useful about the use of language but admits that this would depend on "what is happening on the day and how busy I am". She believes that many reviewers and editors think that it is part of their role to protect academic English and says that this is part of a "slightly hidden agenda [or rather] something they don't really recognise themselves". She feels that this aspect of the reviewing role needs to be openly acknowledged, and that people need to be prepared to justify their defence of current academic English standards.

A number of interesting points were raised in the interviews. The participants appreciate that in the current environment in higher education where English is accepted as the academic lingua franca and academics are strongly encouraged to publish in English medium journals, non-anglophone academics face an ongoing challenge. As indicated, the majority were willing to help non-anglophone writers. Most of the reviewers felt that it was part of their role to identify language errors and point them out to authors. Whether this identification extended to suggestions and advice was largely dependent on the amount of time reviewers had available and how complicated the language issues were. Whether a reviewer acted as an Ally or a Ringmaster, therefore, often depended on time constraints.

The identification of errors extended to nonconventional use of language which was not necessarily ungrammatical or ambiguous. A number of reviewers compared non-anglophone authors to students. It was implied that they regarded these authors in much the same light as they would promising students who needed a little help. They indicated that they attempt to keep the comments constructive. While one appreciates the good intentions behind these approaches, they are nevertheless disturbing. If reviewers position their non-anglophone colleagues as promising students, the attitude is somewhat patronising.

Two interviewees did not adopt this approach. R1 was quite open that she viewed non-anglophone academics as competing with her for publication. It is easy, on the one hand, to condemn her as being self-serving and taking unfair advantage of non-anglophone writers. It is, however, quite possible, that she is merely expressing a view held by many anglophone researchers who are perhaps not quite

as honest as she is. In addition, she pays non-anglophone academics the compliment of viewing them as quite as capable as she is of producing publishable research. R2, who was not prepared to give language advice, believes that some of the best writers in his field were non-anglophones.

Another area of interest was these reviewers' attitudes towards involving anglophone academics in assisting their non-anglophone colleagues with the language of their manuscripts. Boncaegra-Vallle [15] points out that a common recommendation is that non-anglophone authors consult native speakers, the "go-native recommendation", which she argues appears to indicate that many reviewers still believe that "native-like English is the standard to be followed and a baseline for the disciplinary community" (p. 226). The reviewers in this case did not follow the expected trend. Two (R1, R5) recommended that authors consult their anglophone colleagues, two (R3, R4) were noncommittal. and four (R2, R6, R7, R8) were hesitant to make the recommendation. However, three of these four had pragmatic reasons for their hesitation. They were concerned about the cost factor, pointing out that authors might not have access to native speakers who would be willing to help them and that professional editing help was expensive. They were also concerned that the anglophone academics might not have sufficient knowledge of the context to be able to offer sensible advice. There was only one interviewee (R8) who adopted a more ideological position. She argued that simply being an English speaker did not necessarily equip someone to give good advice. "I don't think being a native speaker equates to strong grammar or syntax". This sentiment appears to echo Flowerdew's argument that the practice of asking non-native writers to submit their text to native speakers is "demeaning" ([32], p. 254).

## 5. Conclusions

The subjective nature of the reviewing process is clearly illustrated in the interviews. Acceptable English, it appears, is what the reviewer likes. We need to consider how much of an advantage it is to write in a way that is familiar to the anglophone editors and reviewers of many of our journals. Research indicates that academics can be very emotional in their defence of English as if there is some "purity" at stake ([33], p. 60). The problem with this approach is that it appears to place anglophone reviewers on the moral high ground simply protecting the 'integrity' of an English language style, which happens to be the language with which they are familiar. We need to ensure that the language in the journals that we review and edit adequately conveys meaning. We are entitled to ask for clarity and succinctness—we are not entitled to ask that everyone write American and British English. Perhaps the way forward is for journals to spell out their language policies instead of hiding behind vague terms such as good English/natural English/Standard English. If the journal requires British or American English, they need to acknowledge this upfront. If language requirements are more carefully spelt out, then there is something that can be challenged. At the moment, there is nothing but vague terminology. If journals indicate their stand clearly with regard to the language they find acceptable, it will then be the decision of academics, both non-anglophone and anglophone alike, as to whether they choose to publish in a particular journal.

**Limitations:** This was a very small study that involved only eight participants drawn from two discipline fields. All the participants work at the same university, and the discipline areas have significant overlap, in that applied linguistics involves the teaching and learning of languages. It might be interesting in the future to explore the views of participants from other universities working in different fields.

**Funding:** This research received no external funding.

**Conflicts of Interest:** The author declares no conflict of interest.

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
