# Peer review of "Shakespeare and the English Poets: The Influence of Native Speaking English Reviewers on the Acceptance of Journal Articles"

_publications, doi:10.3390/publications7010020_

Round 1

Reviewer 1 Report

This is a very interesting article which explores the views of eight Anglophone (English-native) reviewers in the fields of Linguistics and Education towards the English language used in research articles submitted for publication in academic journals. The article, however, requires major changes and should not be accepted for publication as is. My main objections to the acceptance of this article refer to the literature review and the methodology and discussion sections.

Literature review is weak and not well presented. Some literature review is subsumed into the introduction but there is no proper section. Here, references to Paltridge prevail (I find it very correct to refer to Paltridge’s works), but I miss some other relevant references that deal with the language of reviewing/review reports and would definitely help to contextualize this study and highlight a research niche. The introduction should introduce the topic and outline the main purpose and structure of the paper. Authors are encouraged to read and refer critically to the works below:

-          Bocanegra-Valle, A. (2015). “Peer reviewers’ recommendations for language improvements in research writing”. In R. Plo Alastrué and C. Pérez-Llantada (eds), English as a Scientific and Research Language. Debates and Discourses. English in Europe series vol.2. Berlin: De Gruyter Mouton, 207-230.

-          Englander, K. & G. López-Bonilla (2011). “Acknowledging or denying membership: Reviewer’s responses to non-anglophone scientists’ manuscripts”. Discourse Studies, 13: 395-416.

-          Fortanet-Gómez, I. & M.F. Ruiz-Garrido (2010). “Interacting with the research article author: Metadiscourse in referee reports”. In R. Lorés-Sanz, P. Mur-Dueñas and E. Lafuente-Millán (eds.), Constructing Interpersonality. Multiple Perspectives on Written Academic Genres. Newcastle-Upon-Tyne: Cambridge Scholars, 243-254.

-          Gosden, H. (2003). “‘Why not give us the full story?’: functions of referees’ comments in peer reviews of scientific research papers”. Journal of English for Academic Purposes, 2: 87-101.

-          Hewings, M. (2006). “English language standards in academic articles: Attitudes of peer reviewers”. Revista Canaria de Estudios Ingleses, 53: 47-62.

The second section, “Methodology”, is the weakest in clarity and needs to be extended, reorganized and rewritten from beginning to end. This section should detail who the participants are and also the procedure that was followed to collect the data (so that the research can be replicated). It should also be divided into subsections so that information is clearly presented (aims or research questions, participants, data collection method and procedure). Because there are no clear specific research questions (even framed as research purposes – perhaps “to investigate the attitudes of journal reviewers who are native speakers of English”, see abstract), the overall purpose of the article appears somewhat confused.

Too many questions need to be answered here about the interviews, the participants and the data collection. How long were the interviews? Were they recorded and transcribed or were themes identified from notes taken during the interviews? Age and sex of participants? Position and experience? Why were these eight participants selected? I was wondering whether participants were native-speakers working in those English-speaking countries (New Zealand, South Africa and Canada), and on the very last paragraph of the whole article I read that they all worked at the same university. These details should have been provided before, when describing the participants. Also, average number of articles reviewed per year? For which journals do they review, international, domestic or both, highly-ranked or not necessarily? Do reviewers speak other languages? Do they review in those other languages? Do all reviewers review for articles in both disciplines or how are they distributed? How the data were analysed is also unclear: what type of coding was used to theme the data? Inductive, deductive, axial, open? Was coding done manually or did authors use any qualitative data analysis software? Did authors themselves do the coding? Was this coding supervised by somebody else –an outside rater? Was the interview piloted at an initial stage?

Section “Findings and discussion” provides results but lacks the breadth and depth expected for a discussion section. In fact, discussion of findings is absent here. I would recommend authors to structure this section into thematic (sub-)headings but, most importantly, to refer and integrate the findings to the literature. By way of example, just the last subsection (Using native speakers of English who where…) contains a couple of references but, particularly the latter (Lillis and Curry) is not discussed against this study, but just outlined. We really do not need that; authors need to refer to literature to support and strengthen their own findings.

I have another series of comments that I feel need clarification:

-          What do authors understand when they talk about “native speakers of English” and “non-native speakers of English”? Current literature refers to “Anglophone writers” and “non-Anglophone/multilingual/plurilingual writers” instead and the use of “native vs non-native” is often avoided or mainly used by “native writers”.

-          Page 6, lines 253-266, this finding is fascinating and authors should dig deeper into it. So are English native reviewers recognizing that, as researchers and writers, they hold a privileged position in academia because of their language? Are they implying that non-native writers are at a clear disadvantage because of that and therefore need training whereas they do not because of their native condition (nativeness)?

-          What is the “English of the Western nations” (page 5)? Please, explain.

-          Page 3: The reference to the permission granted by the ethics committee should be provided as a foot or final note, but not in the main text.

-          Interview excerpts should be somehow identified (with a code or participant pseudonyms) so that readers can make comparisons and the text becomes more reader-friendly. I find it quite annoying when I read “one of the reviewers noted” “another reviewer said”; perhaps it might be the same reviewer all the time, and we cannot identify if that comment was made by a New Zealander or Canadian or whoever.

-          The text is well written but some typos need to be corrected (page 5= final decision to be the editors. > the editors’. / the that I lord over? / Page 7 should at all costs … if at all possible).

-          Use “academics”, “researchers” or “scholars”, but not “people”, to refer to people from academia.

-          Page 7, line 266, one interviewee talks about “the great white majority”. What is that? Who is s/he referring to? Will this be the South African reviewer? As we do not know much about the participants (see my comments above) we cannot contextualize his/her words.

-          What are the implications of this study (if any) for novice and experienced reviewers? And for academic writing and research publication in English-medium journals?

Author Response

Thank you for your comments which have been very useful in the reworking of this article.

The introduction has been altered and the literature review completely rewritten, including all the references suggested by the reviewer. As far as the methodology section is concerned it has been considerably expanded, and I have attempted to address all the issues raised. However, asking for demographic data relating to age is not appropriate in my context. I was the sole author so did all the coding on my own. Having read the suggested articles I revisited my analysis and decided to use the categories suggested by Englander and Lopez-Bonilla (2011). The reference to ethics consent is now a footnote.

The findings and discussion section have been completely rewritten. I have replaced native/non-native speakers of English with the terms anglophone/non-Anglophone. The findings referred to has been enlarged on. All the participants are now identified.

The other matters that the reviewer mentions have been addressed in the overall reworking of the text.

Reviewer 2 Report

This paper reports an interesting qualitative study that investigates English native-speaking reviewers’ attitudes to non-native speaking authors’ articles. The findings reveal a certain degree of bias against non-native speaking authors in international publishing. However, I have some questions. First, I find the literature review separated from the research questions. The literature review focuses on the issues of the challenges that reviewers face as well as the necessity to provide reviewer training. But the research questions are concerned with the reviewers’ attitudes towards the type of English they see in reviewing manuscripts. I can’t see a clear connection between these two parts, and therefore, I think the theoretical significance of the current study has not yet been clearly identified. Second, as a qualitative study, the methods adopted in data analysis are too simplistic. The author has not clearly stated how the different themes of the data emerged from the qualitative data analysis. In other words, the author may want to improve the transparency of his data analysis and the trustworthiness of this piece of qualitative study. Thirdly, although the themes are interesting, I still find the analysis a bit too descriptive. The study mainly describes different participants’ attitudes, without further probing into the implicit mechanism. The author did quote Lillis and Curry’s idea of “centripetal pull”, but I’m afraid it’s not enough. I would like to see much in-depth discussion of the core issues that the findings may reflect, such as the issue of linguistic injustice. Hyland’s 2016 article and the ensuing response articles should also be accounted for in this NES vs. NNES debate. Also, I would recommend another article just published on-line: Hanauer, D., Sheridan, C., and Englander, K. 2019. Linguistic Injustice in the Writing of Research Articles in English as a Second Language: Data From Taiwanese and Mexican Researchers. Written Communication. 36(1) 136–154

Author Response

Thank you for your insightful comments. The literature review has been completely rewritten. The analysis of the themes has also been revisited and considerably tightened. The debate surrounding the issue of linguistic injustice has been covered. I have made extensive use of the article by Hanauer et al 2019.

Round 2

Reviewer 1 Report

Congratulations to the authors after this thorough revision! The paper looks great and has been very much improved. However, I feel that the text has been carelessly edited. I've added just some notes to the pdf file to illustrate what I mean. 

One important issue that needs to be revised in depth: the number of references in brackets within the text do not always match the correct references in the main list of references. I've pointed out some inconsistencies. 

The final list of references also needs revision to match the journal guidelines. 

Authors are required to check and revise typos  throughout the whole text -- like use of ' versus ", which is inconsistent; or " end quotes missing for some quotations.

Author Response

Corrected. Thanks for your comments.

Reviewer 2 Report

This is an extensive revision of the previous draft. All the issues I've raised have been sufficiently addressed. Thus, I recommend publication.

Author Response

Thanks for your comments.